# Are mHealth Interventions Effective in Improving the Uptake of Sexual and Reproductive Health Services among Adolescents? A Scoping Review

**DOI:** 10.3390/ijerph21020165

**Published:** 2024-01-31

**Authors:** Nazeema Isaacs, Xolani Ntinga, Thabo Keetsi, Lindelwa Bhembe, Bongumenzi Mthembu, Allanise Cloete, Candice Groenewald

**Affiliations:** 1Human Sciences Research Council, Pretoria 0001, South Africa; xntinga@hsrc.ac.za (X.N.); tkeetsi@hsrc.ac.za (T.K.); bbslmthembu@hsrc.ac.za (B.M.); acloete@hsrc.ac.za (A.C.); cgroenewald@hsrc.ac.za (C.G.); 2Impact Centre, Cape Town 8001, South Africa; 3Centre for Community-Based Research, Durban 4001, South Africa; 4Public Health, Societies and Belonging (PHSB) Division, Cape Town 8001, South Africa; 5Psychology Department, Rhodes University, Grahamstown 8001, South Africa

**Keywords:** sexual reproductive health (SRH), mHealth, interventions, low-middle income countries, high-income countries

## Abstract

Adolescents continue to face challenges to their sexual and reproductive health (SRH) both locally and internationally. Digital technologies such as the Internet, text messaging, and social media are often viewed as valuable tools for disseminating information on SRH. Mobile health, also known as mHealth, is a medical and public health practise that uses these digital technologies to communicate information. The literature has revealed that mHealth interventions have a positive outcome in delivering SRH information to adolescents. This review aimed to synthesise empirical studies that evaluate mHealth interventions and assess the extent to which these mHealth interventions promote sexual and reproductive health outcomes among young people. This scoping review reviewed the literature across four databases, including EBSCOhost, Scopus, Proquest, and Cochrane, and included 12 articles. The findings have shown that mHealth interventions are effective in enhancing sexual and reproductive health (SRH) knowledge and attitudes among young people in both low-middle and high-income countries. However, comprehensive longitudinal studies are necessary to measure the sustainability and long-term influence of mHealth interventions on behaviour. It is recommended that with artificial intelligence (AI) improvements, there is a possible path to bolstering mHealth interventions.

## 1. Introduction

Adolescence is the period of life between childhood and adulthood [1]. Adolescents are defined as people between 10 and 19 years old [2]. Notably, the ages 10 to 14 are among the most critical for human development yet one of the most poorly understood life course stages [1]. Some of the challenges faced by adolescents across the world in relation to their sexual and reproductive health (SRH) and rights include experiencing gender-based violence (GBV), an increase in unwanted and teenage pregnancies, early marriages, unprotected sex because they have difficulty accessing contraceptives and safe abortion, sexually transmitted infections (STIs), and the transmission of HIV [3,4].

Of concern is that globally, there has been a decline in SRH service utilisation in both high-income countries (HICs) and low-middle-income countries (LMICs) [5]. This decline is more apparent in LMICs [6]. Young people in LMICs between the ages of 10 and 24 years have little to no access to SRH education and services due to several reasons, including social stigma, policies, and procedures affecting the provision of abortion and family planning to girls as well as the attitudes and behaviours of healthcare professionals towards adolescents [7]. Due to the lack of equal access to quality health services and SRH education programmes, young people’s health deteriorates [4]. Thus, to maintain good SRH, adolescents need access to accurate and accessible SRH information and services. Confidentiality, minimal time waiting, and affordable and adolescent-friendly access primarily influence adolescents’ choice of SRH services [8].

Digital technologies such as the Internet, text messaging, and social media provide the tools to disseminate SRH information as communication platforms [9]. Mobile health, or mHealth, is defined as “a medical and public health practise supported by mobile phones, tablets, patient monitoring devices, personal digital assistants (PDAs), and other wireless devices” [10]. A study conducted revealed that a technology-based intervention with adolescents and young adults to improve access to SRH information had an impact on condom use, abstinence, and increased knowledge of STIs and pregnancy among users [11]. Importantly, mHealth interventions have the potential to offer adolescents accurate and non-judgmental SRH information and services [12]. An increased understanding of STIs and pregnancy shows that technology-based interventions already seem to deliver positive outcomes. Low-cost technologies such as Short Message Services (SMSs) and WhatsApp (a social networking application) have been identified as improving SRH knowledge and awareness among adolescents [9]. Hence, our paper aims to consolidate literature that focuses on answering whether mHealth interventions are effective in improving SRH outcomes among adolescents to promote the uptake of SRH globally.

## 2. Methods

### 2.1. Search Strategy

This scoping review followed the PRISMA-SCR “Tip sheets” [13] in Figure 1. The databases and platforms we chose are reliable and reputable for facilitating scoping reviews. We included the following databases and platforms: EBSCOHost, Scopus, Proquest, and Cochrane. These databases and platforms were used to search, identify, and select relevant peer-reviewed articles. The search strategy focused on technology-supported interventions that promote adolescent sexual and reproductive health. The search was worldwide. Four reviewers led the search and screening procedures (Xolani, Nazeema, Allanise, and Candice), and Candice developed the search strategy. The research question this scoping review wants to answer is: Are mHealth interventions effective in improving the uptake of sexual and reproductive health services among adolescents?

### 2.2. Search Terms and Inclusion and Exclusion Criteria

The inclusion and exclusion criteria and the keywords used in this scoping review are provided in Table 1. The reviewers had a ten-year timeline from 2012 to 2022 and only included English-language articles related to the topic under study.

### 2.3. Screening Procedures

The first round of literature searches yielded a total of 636 titles. After removing duplicates and unrelated titles, only 126 articles were screened. The abstracts of 126 papers were reviewed for inclusion, and only 53 met the criteria. The 53 articles were divided among four reviewers (Xolani, Nazeema, Allanise, and Candice), who screened the articles to identify full-text articles for review. Xolani and Nazeema screened 28 identified titles, and Allanise and Candice screened 25 articles. Once the authors completed the screening, each pair of reviewers met to resolve any discrepancies in the screening. Only 26 full-text articles from the first round of the literature search were included for coding. The second round of literature searches yielded 125 titles. After removing duplicates, the number decreased to 117 articles. Abstracts of the 117 articles were reviewed, and only 114 were selected for a full-text review. The 114 articles were divided among four reviewers (Xolani, Nazeema, Allanise, and Candice) to identify full-text articles for review. Xolani and Nazeema screened 54 of the identified titles, and Allanise and Candice screened 60 articles. Only 21 full-text articles from the second literature search were included for coding. A total of 47 articles were included in the final code. Upon conclusion of the coding activity, the final list included 12 papers, as described in Table 2 below. The primary articles were divided among three authors (Thabo, Lindelwa, and Bongumenzi) for full-text coding. Authors Nazeema, Candice, and Allanise were available to resolve coding inconsistencies. Included studies were assessed for eligibility based on population relevance, and all studies must have evaluated an SRH mHealth intervention and not merely described it. This links to the authors’ interest in understanding the value of mHealth interventions to promote positive SRH outcomes among youth. The review only included peer-reviewed journal articles because they will likely have been evaluated for quality, reliability, and validity during peer review.

### 2.4. Coding and Analysis

ATLAS.ti was used to organise and code the selected literature. Adopting a deductive but iterative approach, we developed a set of a priori codes and refined them throughout the coding process to capture the evolving complexity and unanticipated nuances in the literature. Priori codes focused on SRH topics (i.e., contraceptives, pregnancy, abortion, services, etc.), intervention aim, age of participants, gender of participants, country of study, and date of publication (see Table 2). Thematic analysis was thus applied to “identify patterns and themes, done iteratively through referring to primary data sources where needed, including gaps and silences in the literature” [14].

A close analysis of the studies in this review directed us to subsequently adopt the KAPB model to thematically cluster the primary intervention outcomes of these studies, which has also been applied in other scoping reviews on adolescent-focused interventions (see [15]).

## 3. Results

This review aimed to synthesise empirical studies that evaluate mHealth interventions and assess the extent to which these mHealth interventions promote sexual and reproductive health outcomes among young people. Twelve (n = 12) papers were included in this review, and topics covered across the studies are presented in Figure 2, where issues on contraception, pregnancy, sexual relationships, STIs, and HIV emerged as the focal topics. As shown in Table 2, studies generally included adolescent and youth samples between the ages of 13 and 29. Most papers included female participants only, followed by mixed-gender samples. None of the papers included in this review focused on lesbian, gay, bisexual, transgender, intersex, or queer (LGBTIQ+) youth. Most studies adopted rigorous evaluation designs, including randomised control trials (RCTs). More information can be found under the quality assessment of the studies. 

As displayed in Figure 3, all the studies were directed towards enhancing SRH knowledge (n = 12) as a key outcome, while a few also focused on promoting positive SRH behaviours (n = 6), attitudes (n = 2), and practises (n = 2). The next section of this paper will describe our findings concerning KAPB, highlighting differences we observed between LMIC and HIC.

### 3.1. SRH Knowledge

Knowledge enhancement generally entailed the delivery of SRH information related to abstinence, contraceptive methods (e.g., ‘the pill’, IUD, condoms), pregnancy, and sexual relationships. Knowledge enhancement was a key priority in all the studies, and when compared to attitudes, behaviours and practices, it appeared that mHealth interventions proved valuable in improving SRH knowledge. For example, a longitudinal cluster–randomised controlled trial was conducted to evaluate whether reproductive health outcomes (knowledge, self-reported pregnancy, and sexual behaviour) can be improved through text-messaging programmes among adolescent girls aged 14 to 24 years in Accra, Ghana [16]. Their 12-week intervention included three arms, namely a unidirectional intervention (receiving text messages with RH information), an interactive intervention (receiving text messages and RH quizzes), and a control group (placebo with information on malaria). Text message content focused on pregnancy prevention, including “reproductive anatomy, pregnancy, sexually transmitted infections (STIs), and contraception, including male and female condoms, birth control pills, and emergency contraception” [16]. Evaluations were conducted at three months and again at 15 months. At three months, results showed an increase in knowledge for both intervention groups (11% (95% confidence interval [CI] = 7, 15) for the unidirectional intervention and 24% (95% CI = 19, 28) for the interactive intervention). The same intervention was reported to have positive outcomes in increasing adolescent girls’ SRH knowledge [17].

Positive changes in SRH knowledge associated with mHealth interventions have also been found in other LMICs, such as Nicaragua, Indonesia, and Peru. The interactive intervention developed by TeenSmart International for rural and marginalised Nicaraguan youth has shown promising results [18]. This short-term e-learning programme included quizzes, videos, and infographics to improve SRH knowledge, behaviours, skills, and motivations. It was evaluated through a pilot randomised field study and included adolescents aged 14–17 years, and content was delivered through email and an app. Youth in the intervention group scored 8.1% higher on SRH knowledge at the endline compared to those in the control group, with specific improvements observed in knowledge related to STIs, condom use, pregnancy, abstinence, AIDS, and improper relationships [18]. Researchers in Indonesia evaluated the feasibility and acceptability of an SMS intervention aimed at improving young people’s SRH knowledge and knowledge associated with smoking harms [19]. The sample included youth aged 16 to 24 years, males and females, and the intervention was delivered over 10 weeks. The intervention was acceptable and feasible to improve the participants’ SRH knowledge, reporting that the mean score for SRH knowledge significantly increased between baseline and endline (2.7 (95% CI 2.47, 2.94) vs. 3.4 (95% CI 2.99, 3.81) (*p* ≤ 0.01)) [19].

The value of mHealth interventions to produce favourable SRH knowledge outcomes among youth and adolescents living with HIV has also been reported in the literature. In Nigeria, researchers evaluated Social Media to promote Adherence and Retention in Treatment (SMART) intervention through an unblinded randomised controlled trial with youth living with HIV aged 15–24 years [20]. SMART functions as a structured support group that is delivered via social media platforms and aims to enhance HIV knowledge, social support, retention in HIV care, and ART adherence [20]. Although no significant differences in treatment retention were observed between the two sample groups, results showed that the intervention group performed significantly better in HIV knowledge than the control group at the endline. In an urban township in KwaZulu-Natal, South Africa, the InTSHA intervention, a WhatsApp-based programme developed to influence SRH knowledge among adolescents with perinatally acquired HIV, has also shown favourable results [21]. Qualitative interviews with participants aged 15 to 19 who received the Interactive Transition Support for Adolescents Living with HIV (InTSHA) intervention revealed positive feedback on the programme. Specifically, participants described having better SRH knowledge about sex. For some, the private chat forum enhanced their confidence to talk about such sensitive issues with more openness, particularly with their peers who were also living with HIV [21].

Across all the literature in this review, only one study from the LMIC context reported mixed and somewhat unfavourable results about the impact of mHealth interventions on adolescent SRH knowledge. SKILLZ Street is a life-skills school-based programme that uses sport as a platform to empower girls, advocate for their rights, and create a safe space for discussions and learning about girlhood [22]. It includes a two-way SMS campaign called Coach Tumi, where quizzes on topics related to relationships, girlhood, rights, and responsibilities are provided and links to health services that can be accessed in their communities are facilitated. Mixed-methods assessment of the programme showed that upon conclusion of the intervention, there was a 14% increase in the number of participants who knew where to access support services for rape, while there was a decrease in knowledge on where to access other services like pregnancy prevention, HIV testing, and abortion. Interestingly, however, in their qualitative work, authors found that participants expressed SRH knowledge improvements, including how to prevent unwanted pregnancies and HIV infection through abstinence or condom use [22]. Some participants also recounted gaining knowledge and understanding of the “importance of supporting people living with HIV”. In this regard, the intervention did not seem to enhance SRHs knowledge of where to access services. Still, it seemed to have improved knowledge about SRH protective behaviours among a smaller sample group.

Among the HICs, several articles reported increased knowledge of SRH using mHealth interventions. The HEART is a web-based intervention focused on developing sexual assertiveness skills and enhancing sexual decision-making [23]. This intervention recruited participants from four rural, low-income high schools in the southeastern United States of America (USA) [23,24]. Two papers reflect on the HEART intervention [23,24], and in 2017, 222 10th-grade girls in schools participated in the intervention. The study revealed that users found the web programme highly engaging and acceptable [23]. Approximately 90% of girls reported they would recommend the programme to a friend [23]. At a 4-month follow-up, the girls who completed the HEART programme, a new 45 min digital sexual health programme, showed positive results as they had an increase in knowledge on the topic of HIV and other STDs. They also showed more positive attitudes toward condom use compared with girls in the control group [24]. Based on these results, this programme is beneficial for young people to engage with prevention messages [23].

The Teens in New York City (NYC) mobile phone app was created in 2013 as an extension of a paper-based resource. It was launched in 2007 on best practises in sexual health care for adolescents aged 12 to 19 [25]. This app comprises three main parts: Where to Go, What to Get, and What to Expect. Firstly, the Where to Go section focused on assisting adolescents to obtain providers who offer sexual health services or contraceptive methods within their area [25]. Secondly, the What to Get section briefly described each contraceptive method, and the What to Expect section featured videos on sexual health care [25]. The results showed that more than 20,000 adolescents downloaded that app over three years, and more than 28,000 adolescents searched where they should go to obtain these services [25]. Among all 28,000 searches, about 58% selected at least one contraceptive method [25]. The results showed that the mobile app provided adolescents with information on the wide range of sexual health services available in New York and where to access them.

The Healthy U is an app-based, self-paced fatherhood prevention programme that targets male youth, ages 14–19, recruited from the Oregon Youth Authority Juvenile Justices facilities in the USA [26]. This app aims to educate adolescent males on the following topics: Puberty, Pregnancy, Birth Control, HIV, STDs, Healthy Relationships, and Condom Negotiation [26]. For this study, the Healthy U app was administered on an Android tablet through a web-based application. These male youth participated in baseline surveys before treatment group participation in Healthy U and completed surveys six months post-baseline. Healthy U was explicitly designed to increase sexual health attitudes, knowledge, and intentions toward condom and birth control use among male youth. The results revealed that no significant differences were found in knowledge about condom use [26]. Therefore, it is argued that further evaluation is needed to determine the effects of the intervention on a long-term basis, as the results only reflect the first six months.

The miPlan app was developed to provide adolescent girls and young women with information about sexual and reproductive health both inside and outside of the healthcare setting. This app aims to include women who require contraceptive counselling interventions. The miPlan app in family planning clinics among 110 African-American and Latina adolescent girls and young women aged 15 to 24 in the USA was evaluated for its acceptability [27]. They found that nearly 9 in 10 young women reported they would use an app outside of a clinic setting [27]. In addition, participants rated the app as highly acceptable (95.4%) and found it both easy to use and highly informative (97.3%) [27]. Furthermore, 96.4% of the participants reported that miPlan can be used while waiting for their clinic visit [27]. Although this app was highly acceptable and informative, participants reported no change in their self-efficacy for contraception over the study period [27].

A new media intervention called the Sexual Health and Youth (SHY) SMS project aimed to increase knowledge, create awareness, and uptake of sexual health care seeking and sexual health protective behaviours among 119 young people (aged 15–25 years) in Central West Gippsland, Australia [28]. However, only 41 individuals completed the pre-intervention survey, and 20 completed the post-intervention survey. This showed a low reach and low participation in the intervention. Thus, there was a need to evaluate the impact of this intervention, and they discovered the need for improved research translation for new media interventions outside of research settings [28].

Overall, the findings of this review suggest that mHealth interventions are valuable in improving SRH knowledge among adolescent and youth participants across different contexts, despite some interventions showing unfavourable results.

### 3.2. SRH Attitudes

Only two articles reported on attitude change, both conducted in the USA, and both focused on contraceptives, specifically condom use. The Healthy U app, focused on male youth between the ages of 14–19 in the USA, revealed a positive impact on increasing young males’ attitudes and intentions toward condom use and birth control [26]. This was the only study focused solely on males. These impacts at six months post-baseline ranged from 0.34 to 0.46. This study was considered an important first step in demonstrating the effectiveness of Healthy U in increasing attitudes and intentions related to unplanned teen pregnancy among the traditionally underserved population of justice-involved male youth [26].

The HEART programme focused on developing sexual assertiveness skills and enhancing sexual decision-making in adolescent girls in the southeastern USA [24]. The findings found that adolescents showed changes in their attitudes and self-efficacy after briefly engaging with the online intervention [24]. In addition, participants also reported significant differences in changes in their attitudes toward condom use.

### 3.3. SRH Behaviours

Among the LMIC studies, only a few articles (n = 3) assessed the impact of the respective mHealth interventions on behavioural outcomes. In addition to evaluating SRH knowledge, the authors considered the effect of their mHealth intervention on self-reported contraceptive use, pregnancy, and sexual activity [16]. At 15 months, the authors found mixed results where the intervention had no impact on ever having had sex, having had sex in the past year, or reporting pregnancy in the past year in the entire sample [16]. Results further showed that among intervention group participants who have had sex in the past year, both the unidirectional (i.e., receiving text messages with SRH information) and the interactive (i.e., receiving SRH text messages and SRH quizzes) components of the intervention significantly reduced the likelihood of self-reported pregnancy by 86% in the adjusted models (odds ratio [OR] = 0.14; 95% CI = 0.03, 0.71) and 85% (OR = 0.15; 95% CI = 0.03, 0.86), respectively. The interactive intervention was also associated with an increased likelihood of contraceptive use (birth control pills) and a decreased likelihood of emergency contraceptive use. An unexpected result was that the interactive intervention appeared to enhance the risk of condomless sex in the past year (OR = 3.47; 95% CI = 1.12, 10.74). An explanation for this finding offered by the authors was that it could be indicative of a behavioural shift towards using contraceptives to avoid pregnancy rather than condoms [16].

Other literature reported mixed results. For example, in their study with Nicaraguan youth described earlier, it was reported that although significant SRH knowledge improvements were found, no significant behavioural changes in condom use were observed [18]. However, the intervention was significantly associated with a delay in the initiation of sexual intercourse by 29 days [18]. Behavioural outcomes were also assessed in one of the studies that focused on adolescents living with HIV [18]. With a key interest in “improve treatment retention reported that among YLHIV by improving HIV-related knowledge and social support”, the authors found that the SMART Connections intervention did not substantially improve retention or access to social support [20].

Similar to the LMIC results, few studies (n = 2) in HICs reported on behavioural outcomes concerning SRH mHealth interventions. The HEART mHealth intervention results revealed that girls who completed the HEART programme demonstrated more effective behavioural skills in sexual assertiveness and higher self-reported assertiveness [24]. For example, both groups who completed the HEART programme demonstrated better sexual assertiveness skills measured with a behavioural task than the control condition [24]. Findings showed that girls who participated in the 45-min digital sexual health program’s ability to refuse unwanted sexual activity and negotiate condom use improved. It is recommended that future research expand the HEART intervention and evaluate programme impacts on behaviour change over a longer period of time.

The SHY programme used SMS messaging to raise awareness of sexual health as a way to respond to the increasing rates of chlamydia and unplanned pregnancies among 15–25-year-olds in Australia. The study findings show no significant differences in STI testing or sexual risk behaviours between the pre- and post-intervention surveys of the SHY project [28]. Similarly, the miPlan app was designed to operationalise behaviour change theory among adolescents aged 15 to 24 in the USA. It reported no change regarding their self-efficacy for contraception over the study period [27].

In summary, fewer studies considered the behavioural outcomes associated with SHR mHealth interventions, and of those studies that did, behavioural outcomes were difficult to achieve.

### 3.4. SRH Practices

SRH practises entailed the extent to which the intervention encourages youth health-seeking practices. While one study in the LMIC group aimed to improve knowledge of where SRH services can be accessed [22], the researchers did not evaluate whether services were [22], and the researchers did not assess whether services were visited or used by the sample. Likewise, the Teens in New York City (NYC) mobile phone app was created in 2013 and listed healthcare providers in NYC that met best practises in sexual health care for adolescents between the ages of 12 and 19. The app had three main sections: Where to Go, What to Get, and What to Expect. Firstly, the *Where to Go* section of the Teens in NYC app allowed users to find providers of sexual health services or contraceptive methods. Secondly, the What to Get section briefly described each contraceptive method, and the What to Expect section featured videos [25]. The searches revealed that the youth looked for the following services: birth control services, which were used more than 12,000 times, followed by STI testing and treatment, which were used more than 8000 times. Pregnancy testing and abortion services were searched more than 7000 times; HIV testing was searched more than 3000 times; mental health counselling was searched more than 1600 times; and the least explored service was for LGBTQ-specific services, with more than 800 times [25]. The authors further reported that more than 50% of the searches related to contraceptives [25]. In this regard, Steinberg et al. [25] concluded that mobile apps with search functionality are essential for delivering sexual health information to adolescents.

### 3.5. Quality Appraisal

The quality appraisal of all the articles included in this review was the final step in our review, using two separate frameworks. Qualitative papers were appraised using Long and colleagues’ CASP [17] framework, while the assessment tools provided by the NIH were used for quantitative papers. Given the spread of quantitative papers, we drew on three separate NIH Assessment Tools [18], namely the Quality Assessment Tool for Observational Cohort and Cross-Sectional Studies, the Quality Assessment of Controlled Intervention Studies, and the Quality Assessment Tool for Before-After (Pre-Post) Studies with No Control Group.

Three of the 12 articles in this review utilised a qualitative methodology, with one mixed-methods study [20,21,22,28]. All four articles presented the aims of the studies, and the qualitative approach applied appeared appropriate to address the descriptive and/or exploratory aims of the research. However, none of the qualitative articles described their theoretical underpinnings, nor did the authors present their positionality within the papers. Only one article reported the ethics approval number, and none outlined the ethical considerations within the paper in detail. Further, qualitative studies generally mentioned the limitations of their findings for generalisability. Thus, to address this restriction, the authors adequately discussed their work in relation to the available literature, providing research and practise implications.

Most (n = 6) of the quantitative articles employed RCT methodologies, with one article utilising a pre-test-post-test design and another adopting a cross-sectional approach [16,18,23,24,25,26,27]. As expected, quantitative articles incorporated larger sample sizes and more structured evaluation approaches. Quantitative studies described their methodologies adequately, employing baseline and endline analyses on the same sample groups. In most studies, the authors obtained their participants through similar contexts (generally through schools) and uniformly applied their inclusion/exclusion criteria during recruitment. Most RCT studies also provided power estimations for their sample size calculations. However, it was difficult to ascertain whether researchers could control for the influence of other interventions; thus, causal inferences on positive and negative outcomes are difficult to establish. It was also challenging to determine attrition rates among the articles, as only two mentioned this, while others only described their findings, often concerning the intervention group. Studies that employed a pre-tests-post-test approach (including RCTs) conducted post-test assessments immediately following intervention exposure, with a few collecting data between 1 month and 15 months post-intervention [16,24].

Overall, the methodologies employed across the qualitative and quantitative studies adequately fit the aims of the studies, although they are not always justified in depth. The structured quantitative assessments produced fewer risks of bias, and a welcomed finding was the availability of RCT studies employing rigorous evaluation approaches. Qualitative studies would benefit from describing their conceptual or theoretical frameworks and methods in more detail to ensure rigour. Still, the findings presented across all studies offered insights into the value of mHealth interventions for adolescent SRH, as described above.

## 4. Discussion

This review of mHealth interventions aimed at improving SRH among adolescents and teens reflects a consistent and significant improvement in knowledge. Several studies, specifically those conducted in LMICs, have shown that mHealth interventions effectively enhance SRH knowledge among young people [16,18,19]. Similarly, in HICs, the girls who completed the HEART programme had statistically significant knowledge regarding HIV and other STIs [24]. Although the miPlan app was highly acceptable and informative, participants reported no change in their self-efficacy for contraception over the study period [27]. The Sexual Health and Youth (SHY) project aims to increase adolescents’ sexual health knowledge. Still, there was low reach and low participation in the intervention [28].

A difficulty arises in translating this increased knowledge into desirable behavioural results. Rokicki et al. [16] reported inconclusive findings; although there was an improvement in knowledge, there were no statistically significant changes in specific behaviours, such as condom use. Comparable trends were noted in several other studies conducted in both LMICs and HICs [24,28]. Several factors contribute to the difficulty in effecting behavioural changes despite improved SRH knowledge. The discrepancy between knowledge acquisition and behaviour modification is often rooted in complex dynamics: adolescent behaviour is impacted by several socioeconomic influences, including peer influence, cultural standards, and family dynamics. Although individuals may possess a greater understanding, existing behavioural patterns may not readily adapt to change due to deeply rooted social and cultural effects [16]. For example, authors proposed a transition towards the use of contraceptives to prevent pregnancy rather than relying on condoms, which may imply a prioritisation of pregnancy prevention over STI risk reduction [16].

Many interventions focus on augmenting knowledge rather than implementing effective behavioural change strategies [24]. Studies emphasising behavioural changes achieved relatively limited success compared to those only aiming for knowledge enhancement [24,28]. Greater focus is necessary for tailoring interventions to incorporate evidence-based behaviour change models, enabling a more sophisticated approach to create long-lasting behavioural changes. Behavioural changes are multifaceted and complex, often requiring longer-term evaluations beyond the scope of many studies [26]. For example, the Healthy U app indicated positive short-term attitudes about contraceptive use among male youth but did not reveal significant knowledge changes [26]. Comprehensive longitudinal studies are necessary to measure the sustainability and long-term influence of mHealth interventions on behaviour.

Some successful interventions, particularly those addressing attitudes and behaviours, offer insights into practical strategies for behaviour modification. The Health Education and Relationship Training (HEART) programme, which focused on developing sexual assertiveness skills, exhibited rapid positive changes in attitudes and assertiveness among adolescent girls [24]. The effectiveness of such programmes underlines the importance of addressing specific behavioural domains. Expanding the scope of activities beyond simply knowledge dissemination is pivotal. Incorporating comprehensive programmes addressing social variables, behavioural models, and cultural concerns might yield more meaningful and sustainable changes [16,24].

The review highlights a scarcity of studies concentrating on male SRH and menstruation concerns. Greater research attention is needed in these areas to increase the range of current studies and ensure inclusion and a holistic understanding of SRH among varied groups. Crafting a comprehensive mHealth programme demands a multi-dimensional approach. Such an intervention should integrate SRH knowledge dissemination with evidence-based behaviour modification models, addressing social variables, cultural norms, and unique demographic needs [21]. Incorporating numerous means of communication, including interactive platforms, quizzes, videos, and forums, provides for targeted involvement and privacy for teenagers [18,21].

SRH is primarily focused on maternal and other reproductive healthcare services that aim to assist heterosexual and cisgender women and often does not respond to the needs of diverse populations, including transgender women [29]. Achieving the goal of advancing the effectiveness of mHealth interventions and improving the uptake of SRH requires an inclusive response. Access to non-judgmental and gender-affirming SRH services is an integral part of a holistic response to healthcare for LGBTQ people [30]. All people who are capable of becoming pregnant, which may include lesbians, trans men, and nonbinary people, need family planning and abortion care [30]. Transgender women’s lack of access to sexual and reproductive health and rights (SRHR) is due to a health system that is unresponsive and governed by pervasive heteronormativity in the design, management, and provision of services [31].

Even though mHealth interventions have the potential to offer adolescents accurate and non-judgmental SRH information and services [30], we found that none of the papers included in this review focused on LGBTIQ+ youth. This is a surprising finding, given that there has been such a dramatic increase in the use of technology for HIV prevention research and practise [12]. The fact that none of the papers included in this review focused on LGBTIQ+ youth draws our attention to the apparent neglect of the particular SRH needs of this population and perhaps also the lack of allocated funding and commitment for SRH mHealth interventions at the public healthcare level that address the specific SRH needs of LGBTIQ+ youth. To promote access to SRH mHealth services while acknowledging inclusivity, it is also crucial that LGBTIQ youth are not seen as a monolithic population [32]. Individuals within the LGBTIQ youth community have different needs, experiences with barriers, and levels of access to care.

In the context of the needs of trans and gender-diverse people, access to gender-affirming healthcare forms part of the SRH response. The SADC Regional Strategy on SRHR includes transgender persons as a key population to advance inclusive SRH responses [32]. However, even though legislation and policies aim to be inclusive, those policies often need to be enacted.

### Challenges and Opportunities for mHealth Interventions

While most studies have described their mHealth intervention as user-friendly, enjoyable, and generally acceptable by adolescent and youth samples, researchers also described the challenges they faced in delivering these interventions. Notably, the value of mHealth interventions to facilitate private conversations has been reported [20,22], yet privacy restrictions have emerged as a primary challenge in other studies. A common concern in digital health interventions is the privacy-related risk to participants, as an outside person (parent/caregiver, partner, friend) may view the messages on the participant’s phone against their wishes. This is particularly problematic when youth do not have their mobile phone and are required to use someone else’s phone to access intervention content (which is generally on sensitive and/or private issues) (see also [18,21,24,25,28]). Technological challenges also emerged as a concern, where participants experienced challenges like the inability to purchase cellphone minutes, difficulties accessing the Internet or messages, a lack of storage space on mobile phones to accommodate the intervention content [18], and slowness in internet connection speed [23]. A further challenge that emerged related to the applicability and depth of the intervention content. Ensuring that intervention content is contextually relevant was an important consideration in Indonesia. The authors found that some participants felt uncomfortable with the selected intervention content that was considered to promote extramarital relationships [19]. Limited depth or ‘unmet health needs’ was also reported as a challenge, where participants concluded the intervention with additional questions related to HIV disclosure, HIV transmission, sexual abuse, age-disparate relationships, abortion, contraception, and communication within relationships [21]. For HICs, most of the interventions took place within urban areas because of internet access and connectivity [25,26,27,28], and only two HIC interventions recruited participants from rural areas [23,24].

Moreover, a prominent and widely recognised challenge associated with mHealth interventions is that they are designed for those with access to mobile phones. This expected limitation is often challenged by the rapidly growing accessibility of mobile phones and other digital technologies in both HIC and LMIC contexts [16,18,19,20,21,22,23,24,25,26,27,28]. In contemporary society, within the artificial intelligence (AI) revolution, the establishment of mHealth interventions may be further advanced through AI, as AI-driven interventions, like chatbots or ChatGPT, are already considered valuable additions to encourage health-seeking behaviour [33]. An imperative is thus to evaluate how such advances could be adapted towards targeted interventions to facilitate improvements in SRH knowledge or information seeking and work with such programmes towards SRH behaviour change. Indeed, as the authors articulate: “Future, scaled-up mHealth interventions can lower SRH stigma by expanding access to sexual education and peer support, supplementing adolescents’ existing SRH education” [21].

The realisation of a complete SRH mHealth programme in contemporary society faces challenges such as privacy concerns, technological barriers, and the necessity for contextual relevance. Overcoming these challenges requires a deliberate effort to provide targeted, contextually relevant, and accessible treatments while negotiating issues related to digital literacy, device access, and privacy concerns [18,28].

With improvements in AI, there is a possible path to bolstering mHealth interventions. AI-driven treatments, such as chatbots or AI-supported educational modules, could be vital in providing tailored, scalable, and accessible solutions for SRH education and behaviour change [18]. These innovations can augment existing programs, enabling more involvement and accessibility. AI can potentially increase the impact of mHealth interventions on SRH for young people. AI systems can analyse user data to provide customised SRH recommendations based on adolescents’ requirements and preferences. The authors argued that personalisation increases engagement and the relevancy of the material, leading to a more effective intervention [34,35]. By utilising machine learning algorithms, mHealth interventions can be directed toward the demographics and regions where they are most needed, ensuring a more targeted and efficient approach [36]. AI is also capable of analysing user behaviour patterns, providing insightful data about how well interventions work. AI-driven solutions can adjust in real-time, adjusting content and strategies to optimise behavioural outcomes [37]. Teenagers may easily use and scale up AI-powered chatbots to get information, advice, and support on sexual and reproductive health. These virtual assistants provide a private, judgment-free forum for debates by involving users in real-time dialogues [38,39].

## 5. Conclusions

In conclusion, mHealth interventions promise to boost SRH knowledge among adolescents, although attaining tangible behavioural changes remains a persistent issue. To realise the full potential of these interventions, addressing the gap between knowledge enhancement and behaviour modification is critical. A holistic approach incorporating diversified SRH subjects, contextually relevant content, technological innovations, and inclusive techniques is vital to bridge this gap and create beneficial behavioural adjustments among young people addressing SRH practices. The continuous evolution of mHealth interventions, an interdisciplinary approach, innovative technological adaptations, and comprehensive evaluation strategies will advance adolescent SRH and ensure the successful uptake of sexual and reproductive health services globally. By harnessing AI capabilities, mHealth interventions can become more tailored, adaptive, and targeted, leading to better SRH outcomes for young people worldwide.

This work is essential because it is crucial to synthesise empirical studies that evaluate SRH mHealth interventions so that we can map best practices, identify the gaps, and make contextually relevant recommendations to improve the uptake of SRH services for young people. Hence, in presenting the currently available evidence, our scoping review advances the science of enhancing the uptake of SRH mHealth interventions for young people by identifying the gaps and highlighting the best practices.

## Figures and Tables

**Figure 1 ijerph-21-00165-f001:**
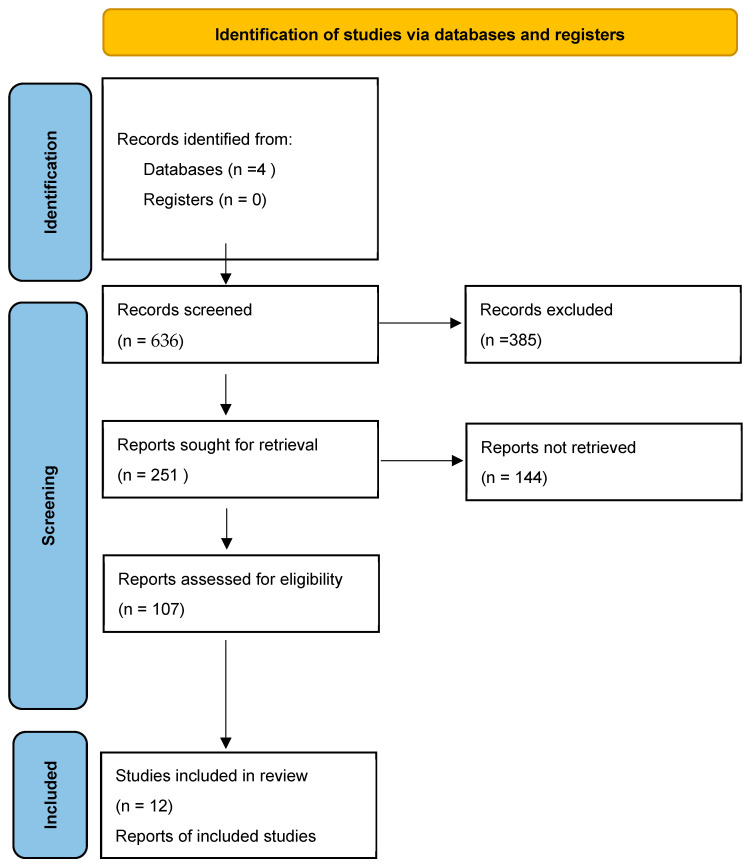
PRISMA flow diagram for selection of eligible studies [13].

**Figure 2 ijerph-21-00165-f002:**
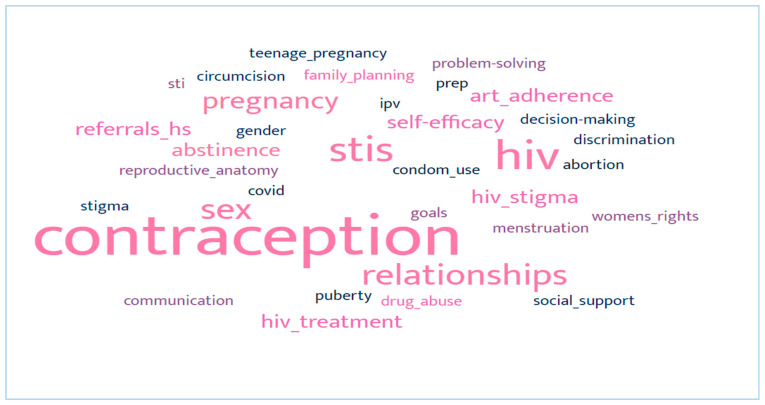
Topics covered across papers.

**Figure 3 ijerph-21-00165-f003:**
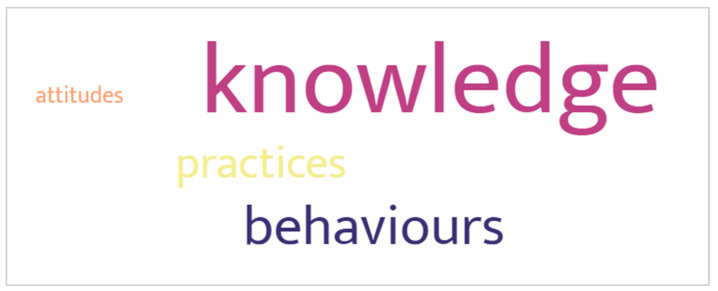
KAPB.

**Table 1 ijerph-21-00165-t001:** Scoping review criteria.

Keywords related to technology	mobile technology OR mobile devices OR cell phones OR tablets OR electronic or online OR chatbot OR virtual agent OR virtual assistant OR bot OR conversational agent OR artificial intelligence mHealth OR m-health OR mobile health OR mobile app OR mobile application OR digital health OR tele-health OR telehealth OR digital technologies OR digital technology OR short messaging service OR SMS OR web-based OR web based OR hotline OR hot line OR hot-line OR unstructured supplementary service data OR USSD OR social media OR social media platform
AND	
Keywords related to sexual and reproductive health	sexual and reproductive health, OR SRH
Keywords related to interventions	Intervention OR strategies OR program OR ‘best practices’ OR trials
Keyword placement	Title
Inclusion criteria	Peer-reviewed articles (qualitative and quantitative) Meta-analysis articles Articles published in English Full-text articles
Exclusion criteria	Commentaries Editorials Books Studies that do not include keywords. Studies that do not explicitly align with the aim of the review. Articles published in a language other than English Grey literature reports and; Non-peer-reviewed conference proceedings

**Table 2 ijerph-21-00165-t002:** Description of the studies.

Title	Authors	Country	Age	Gender	Aim
Impact of a Text-Messaging Program on Adolescent Reproductive Health: A Cluster–Randomised Trial in Ghana	Slawa Rokicki; Jessica Cohen; Joshua A. Salomon; and Gunther Fink,	Ghana	14–24	Female	The aim is to use text messaging to improve adolescent girls’ reproductive health.
A Social Media–Based Support Group for Youth Living with HIV in Nigeria (SMART Connections): A Randomized Controlled Trial	Lisa Dulli, Kathleen Ridgeway, Catherine Packer, Kate R. Murray, Tolulope Mumuni, Kate F. Plourde, Mario Chen, Adesola Olumide, Oladosu Ojengbede, and Donna R. McCarraher	Nigeria	14–24	Mix	The aim was to test the effectiveness of an intervention delivered through a social media platform.
Linking at-risk South African girls to sexual violence and reproductive health services: A mixed-methods assessment of a soccer-based HIV prevention programme and pilot SMS campaign	Katherine G. Merrilla; Jamison C. Merrilla; Rebecca B. Hershowa; Chris Barkleya; Boitumelo Rakosab; Jeff DeCellesa; Abigail Harrison	South Africa (SA)	11–16.	Female	The grassroots soccer community developed SKILLZ Street, which is a soccer-based life skills programme using SMS to support adolescent girls at risk for HIV, violence, and sexual and reproductive health challenges.
I am not shy anymore”: A qualitative study of the role of an interactive mHealth intervention on sexual health knowledge, attitudes, and behaviours of South African adolescents with perinatal HIV.	Scarlett Bergam; Thobekile Sibaya; Nompumelelo Ndlela; Mpume Kuzwayo; Messaline Fomo; Madeleine H. Goldstein; Vincent C. Marconi; Jessica E. Haberer; Moherndran Archary; and Brian C. Zanoni	South Africa	15–19	Mix	Using mHealth interventions delivers information, fosters social support, and improves decision-making skills. In this study, we evaluate how a health intervention influences sexual health knowledge and behaviours in APHIV.
Harnessing the Power of Technology to Improve Sexual and Reproductive Youth Health in Nicaragua: A Randomised Field Study	Adriana Gómez Gómez; Carolina Alfaro González; Catherine Strachan Lindenberg; Sara Benítez Majano; Vilma Medrano García; Luis F. Guillen; Mariana Harnecker Romanjek; and Mary Coffman	Nicaragua	14–17	Not mentioned	This study aimed to test an SRH promotion intervention to increase adolescent knowledge, skills, motivations, and behaviours.
A quasi-experimental text messaging trial to improve adolescent sexual and reproductive health and smoking knowledge in Indonesia	Alisa E. Pedrana; Jamie PinaC; Retna S. Padmawati; Ririh Zuhrina D; Lutfan Lazuardi F; Megan S. C. Lim; Margaret E. Hellard; and Yayi S. Prabandari D,	Indonesia	16–24	Mix	To evaluate the feasibility and acceptability of a text message intervention aiming to improve young people’s knowledge of sexual reproductive health (SRH) and harms related to smoking.
Challenges to translating new media interventions in community practice: a sexual health SMS program case study	Cassandra J. C. Wright; Kaytlyn Leinberger; and Megan S. C. Lim	Australia	15–25	Not mentioned	The translational challenges for new media interventions are illustrated by the Sexual Health and Youth (SHY) program that uses SMS.
Development of a Mobile App on Contraceptive Options for Young African American and Latina Women	Motolani Akinola; Luciana E. Hebert; Brandon J. Hill; Michael Quinn; Jane L. Holl; Amy K. Whitaker; and Melissa L. Gilliam	United States of America (USA)	15–29	Female	The miPlan app was designed, and the acceptability of using this app in family planning clinics was evaluated. The aim was to provide health information due to users’ ability to identify with them and their accessibility.
Feasibility and acceptability of a web-based HIV/STD prevention program for adolescent girls targeting sexual communication skills	L. Widman; C. E. Golin; K. Kamke; J. Massey; and M. J. Prinstein	USA	<18	Female	This study assessed adolescents’ feasibility and acceptability of this new Health Education and Relationship Training (HEART) program.
Impacts of Healthy U: A cluster-randomised control trial of a sexual health education app developed for justice-involved male youth	Staci J. Wendt; Jonathan Nakamoto; Pamela MacDougall; and Anthony Petrosino	USA	14–19	Male	The study investigated the impact of Healthy U, which is a programme designed for male justice-involved youth focusing on fatherhood prevention.
Evaluation of a Mobile Phone App for Providing Adolescents with Sexual and Reproductive Health Information, New York City, 2013–2016	Allyna Steinberg; Marybec Griffin-Tomas; Desiree Abu-Odeh, Mphil; and Alzen Whitten,	USA	12–19.	Mix	The New York City (NYC) Department of Health and Mental Hygiene released the Teens in NYC mobile phone application (app) in 2013 as part of a programme to promote sexual and reproductive health care among adolescents.
Sexual Assertiveness Skills and Sexual Decision-Making in Adolescent Girls: A Randomized Controlled Trial of an Online Program	Laura Widman; Carol E. Golin; Kristyn Kamke; Jeni L. Burnette; and Mitchell J. Prinstein,	USA	Mean age 15.2	Female	To evaluate the efficacy of the HEART app developed for sexual assertiveness skills and enhancing sexual decision-making in adolescent girls.

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
