# Peer review of "Are mHealth Interventions Effective in Improving the Uptake of Sexual and Reproductive Health Services among Adolescents? A Scoping Review"

_ijerph, 2024, doi:10.3390/ijerph21020165_

Round 1
Reviewer 1 Report
Comments and Suggestions for Authors
We thank the authors for addressing such an important public health topic affecting adolescent sexual health globally. However, my major concern is the exclusion of several peer-reviewed articles that are relevant to the research questions.
My comments are the following:
1. For the 2.2. Search terms and inclusion and exclusion criteria, please replace "systematic review" with "scoping review". I believe this was a typo.
2. I believe that some essential studies were excluded from this review despite the relevance to the topic. Please take a look at the following studies as they might add value to your review:
a. Shegog, R., Craig Rushing, S., Gorman, G., Jessen, C., Torres, J., Lane, T. L., Gaston, A., Revels, T. K., Williamson, J., Peskin, M. F., D'Cruz, J., Tortolero, S., & Markham, C. M. (2017). NATIVE-It's Your Game: Adapting a Technology-Based Sexual Health Curriculum for American Indian and Alaska Native youth. The journal of primary prevention, 38(1-2), 27–48. https://doi.org/10.1007/s10935-016-0440-9
b. McCrimmon, J., Widman, L., Javidi, H., Brasileiro, J., & Hurst, J. (2023). Evaluation of a Brief Online Sexual Health Program for Adolescents: A Randomized Controlled Trial. Health promotion practice, 15248399231162379. Advance online publication. https://doi.org/10.1177/15248399231162379
The search strategy across the databases should be carried out once again as this is a major flaw of this paper which led to the exclusion of relevant to research papers. I would also recommend setting a timeframe (time of study publication) when including/excluding articles to facilitate the process.
Author Response
Please see the attachment for the responses to the reviewer's comments.

Reviewer 2 Report
Comments and Suggestions for Authors
Congratulate the authors for the work done, which is very interesting and necessary for the current situation. In relation to work, highlight some aspects of improvement.
1. Unify the format of citations throughout the work, sometimes the authors are mentioned and in others only the numbering corresponding to the references. It is recommended to put the corresponding numbers in references or cite it, according to the journal's standards.
2. It would have been interesting to include the JCR database, being one of the most important and recognized worldwide.
3. When the reviewers of the articles are referred to with the acronyms This clarification is important for readers.
4. The years that have been considered in the search are not indicated in any table or in the text. It is interesting to clarify the reason why the language is not limited either, because if the reviewers are multilingual it is justified, but it is necessary to indicate it in the work.
5. There are tables that are recommended to be reduced in length, such as number 2 being very extensive.
6. In table 5, even if you know what the acronyms Y, N and NA mean, it would be advisable to indicate at the bottom so that all types of readers can understand what they mean.
7. Although several headings are presented in the results, it would be advisable to indicate in the method the research questions that we want to answer with this review.
8. The conclusions would be appropriate to expand the information, they are very synthetic.
9. Review the references that include all consulted sources included.
All of these are minor aspects that are recommended to be reviewed to improve the publication.
Author Response
Please see the attachment of the authors response to the reviewer's comments.

Reviewer 3 Report
Comments and Suggestions for Authors
Review
Congratulation, the topic is original and relevant to the field.
Among the evaluated manuscripts only HEART web-based interventions shows with clarity from where the population sample was recruited. It would be interesting to know if the authors have data in this sense in the other articles.
Also the quality appraisal is a very good tool for evaluating articles, but the tables are hard to follow being too detailed.
We do not know within the included articles the economic-socio-cultural environment from which adolescents come from, this aspect not being evaluated by the authors in this review. Could the authors add some details?
The conclusions are consistent and the references appears relevant to the chosen topic.
Thank you very much.
Author Response

(The authors gave the same response as above.)

Reviewer 4 Report
Comments and Suggestions for Authors
The authors aim to synthesise empirical studies that evaluate mHealth interventions and assess the extent to which these mHealth interventions promote sexual and reproductive health outcomes amongst young people.
In my opinion this study is very interesting and useful. The authors have presented the material well and it is acceptable without revision.
Author Response
Thank you for your positive comments. We the authors appreciate it alot.
Round 2
Reviewer 1 Report
Comments and Suggestions for Authors
We thank the authors for addressing the raised concerns.
Reviewer 2 Report
Comments and Suggestions for Authors
The proposals for improvements to the work have been addressed and I consider that the work has improved compared to the previous version.